# Randomized clinical trial to assess the protective efficacy of a *Plasmodium vivax* CS synthetic vaccine

Myriam Arévalo-Herrera[1,2], Xiomara Gaitán[1], Michelle Larmat-Delgado[1], María Alejandra Caicedo[1], Sonia M. Herrera[2], Juliana Henao-Giraldo[1], Angélica Castellanos[1], Jean-Christophe Devaud[3], André Pannatier[3], José Oñate[4], Giampietro Corradin[5] & Sócrates Herrera [1,2 ✉]

A randomized, double-blind, controlled vaccine clinical trial was conducted to assess, as the primary outcome, the safety and protective efficacy of the *Plasmodium vivax* circumsporozoite (CS) protein in healthy malaria-naïve (phase IIa) and semi-immune (phase IIb) volunteers. Participants (n = 35) were randomly selected from a larger group (n = 121) and further divided into naïve (n = 17) and semi-immune (n = 18) groups and were immunized at months 0, 2, and 6 with *Pv*CS formulated in Montanide ISA-51 adjuvant or placebo (adjuvant alone). Specific antibodies and IFN-γ responses to *Pv*CS were determined as secondary outcome; all experimental volunteers developed specific IgG and IFN-γ. Three months after the last immunization, all participants were subjected to controlled human malaria infection. All naive controls became infected and drastic parasitemia reduction, including sterile protection, developed in several experimental volunteers in phase IIa (6/11) (54%, 95% CI 0.25–0.84) and phase IIb (7/11) (64%, 95% CI 0.35–0.92). However, no difference in parasitemia was observed between the phase IIb experimental and control subgroups. In conclusion, this study demonstrates significant protection in both naïve and semi-immune volunteers, encouraging further *Pv*CS vaccine clinical development. Trial registration number NCT 02083068. This trial was funded by Colciencias (grant 529-2009), NHLBI (grant RHL086488 A), and MVDC/CIV Foundation (grant 2014-1206).

[1] Malaria Vaccine and Drug Development Center (MVDC), Cali, Colombia. [2] Caucaseco Scientific Research Center, Cali, Colombia. [3] Centre Hospitalier Universitaire Vaudois, Lausanne, Switzerland. [4] Centro Médico Imbanaco, Cali, Colombia. [5] Biochemistry Department, University of Lausanne, Epalinges, Switzerland. ✉email: sherrera@inmuno.org

An estimated ~229 million clinical cases and >409,000 deaths occurred worldwide in 2019 due to malaria, with a substantial economic impact on populations living in developing regions[1]. *Plasmodium vivax* is the second species of epidemiological importance with a wide geographical distribution in Asia, Oceania, and the American continents, where it coexists with *P. falciparum*, producing ~6.5 million cases annually[1]. Although it has been historically considered to cause benign disease, *P. vivax* infection results in a very algid febrile syndrome with headache and malaise and frequent pulmonary and hematological manifestations[2,3]. Moreover, severe illnesses, including cerebral malaria and death, have been recently documented[4,5]. Notably, the infection produces hypnozoites, dormant liver parasite forms that relapse periodically, generating clinical manifestations and contributing to an undetermined proportion of the *P. vivax* incidence worldwide[6,7]. Substantial evidence supports the feasibility of developing malaria vaccines, which are considered valuable tools to complement classical malaria control strategies[8]. *P. falciparum* RTS,S/AS01E, based on the circumsporozoite (CS) protein, is currently the most advanced malaria vaccine candidate with a mean efficacy of ~34% in phase III trials[8,9]. Recently the World Health Organization (WHO) recommended widespread use of the RTS,S/AS01 (RTS,S) malaria vaccine among children in sub-Saharan Africa and other regions with moderate to high *P. falciparum* malaria transmission[10]. The recommendation is based on a pilot implementation program ongoing since 2019 in Ghana, Kenya, and Malawi in more than 800,000 children[11]; in addition, several other candidates are in advanced clinical development[12,13]. Although *P. vivax* vaccines have received significantly less attention, a systematic analysis of the *P. vivax* CS antigen (*Pv*CS) has been performed over the last more than 20 years[14], leading to promising results for its development.

After identification of multiple B- and T-cell epitopes[15,16], three long synthetic peptides (LSP) covering the amino-terminal (N), the central repeats (R), and C-terminal regions were designed and synthesized. Studies in BALB/c mice and *Aotus* monkeys showed high LSP immunogenicity[15,17]. Phase I vaccine clinical studies were subsequently conducted in healthy malaria-naive volunteers (*n* = 69) to individually evaluate the three LSP formulated in Montanide ISA-720 (Seppic, Paris, France). A vaccine dose-escalating protocol using 10, 30, and 100 µg/dose[18] indicated good safety, tolerability, and immunogenicity in a phase Ia trial. A second phase I study was conducted in 40 volunteers who were vaccinated three times with different combinations of the peptides formulated in either Montanide ISA-720 or Montanide ISA-51 at 50 and 100 µg/dose[19]. Because of the known immunodominance of the R fragment, it was only included in two of the immunization doses to equilibrate the response to the three protein fragments. Vaccine formulations were well-tolerated, and no serious adverse events (SAE) were observed. All immunized individuals seroconverted and developed comparable ELISA titers of antibodies to the three protein fragments, which also recognized the native protein on *P. vivax* sporozoites as determined by indirect immunofluorescence test (IFAT). Antibodies to the three fragments inhibited sporozoite invasion (ISI) to liver-cell lines in vitro, in the same proportion[20].

Further studies allowed the standardization of a *P. vivax* sporozoite controlled human malaria infection (CHMI) in healthy and semi-immune volunteers[21–23]. *Pv*CS recombinant products were also developed and tested in phase I trials in studies by other groups. In those studies, the R region of the VK210 *P. vivax* variant was expressed in *E. coli*[24], formulated in Alum, and tested in 13 volunteers (doses ranging from 10 to 1000 µg/dose), displaying a good safety profile but low and no boostable ELISA antibody response. A recombinant protein expressing 70% of the *Pv*CS protein sequence was produced in yeast[25], formulated in Alum and

tested in doses ranging from 50 to 400 µg/dose. Volunteers exposed (*n* = 30) to the higher doses (200–400 µg/dose) generated minimal humoral and cellular responses. Then, a hybrid *E. coli* recombinant construct (VMP001) encompassing VK210 and VK247 repeats alleles[26] was expressed, formulated in AS01B adjuvant, and evaluated in a phase I/IIa vaccine trial. Volunteers (*n* = 30) developed robust humoral and T-cell responses, and a slight delay in patency (1–2 days) was observed in 59% of the volunteers.

Based on the reproducible results of the *Pv*CS LSP phase I trials and the establishment of *P. vivax* sporozoite CHMI protocols, a comparative phase IIa/IIb clinical trial was designed and conducted in a malaria-free region of Colombia to evaluate the safety and protective efficacy of *Pv*CS LSP formulated in Montanide ISA-51 adjuvant in healthy malaria-naive (phase IIa) individuals and the feasibility of the same study protocol in semi-immune volunteers (phase IIb).

## Results

**Volunteer enrollment and retention**. Aiming to recruit 52 potential participants to enroll 36 volunteers from malaria-endemic and non-endemic regions, an extensive trial promotion using posters, flyers, and mass media (radio and TV broadcasting) was conducted as in previous clinical studies[23,27]. From a larger group of interviewed subjects, 121 accepted to be screened to participate in the trial (38 naive, 83 semi-immune) (Fig. 1). However, 80 did not meet the inclusion criteria and six of the remaining 41 candidates declined to participate. Therefore, we decided to initiate the study with the 35 eligible individuals

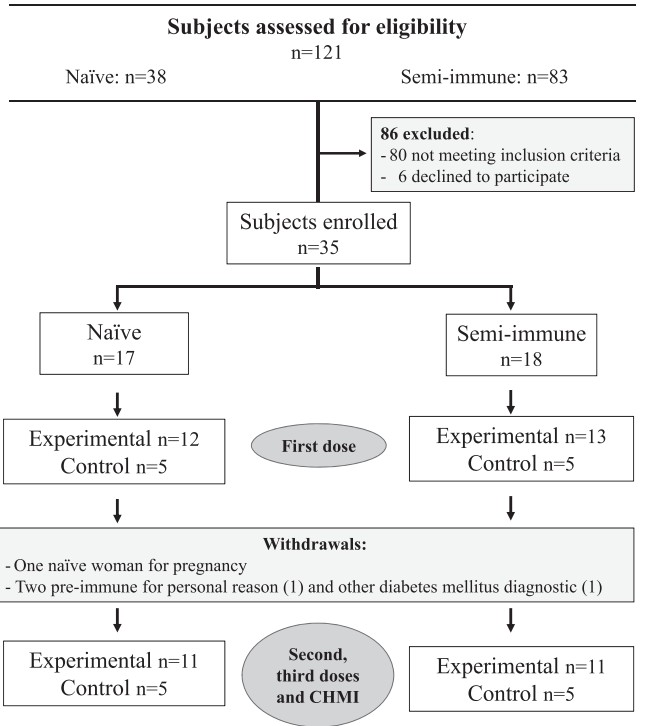

**Fig. 1 Study flow diagram.** The study flow diagram describes the number of individuals in the screening, immunization, and CHMI steps. From 121 volunteers who initially accepted screening (38 naive, 83 semi-immune), 86 were excluded or declined participation and 35 were enrolled. Seventeen were naive and were allocated to Phase IIa study [12 Experimental (Exp) + 5 Control (Ctrl)], and 18 semi-immune to the Phase IIb study (13 Exp + 5 Ctrl). All 35 volunteers (age range 19-44 years) were immunized with *Pv*CS LSP or placebo formulated in Montanide ISA-51. Two volunteers withdrew after the first immunization, and one more (semi-immune) was dropped out because of diabetes mellitus diagnosis.

enrolled. Seventeen naive [12 Experimental (Exp) + 5 Controls (Ctrl)] volunteers were allocated to the Phase IIa group and 18 semi-immune (13 Exp + 5 Ctrl) to the Phase IIb group Table 1. All 35 volunteers were immunized at 0, 2, and 6 months with mixtures of the LSP derived from PvCS (Fig. 2) or with placebo formulated in Montanide ISA-51; a sporozoite CHMI was performed at month 9. Blood samples from volunteers were drawn at times 0, 1, 3, 7, and 10 months for assessment of the immune response (Fig. 3). Two volunteers withdrew after the first immunization (one naive and one semi-immune). One more semi-immune volunteer had to be dropped out because of diabetes mellitus diagnosis, which was considered a SAE not related to the immunization. Therefore, for immunizations 2 and 3 and CHMI, both the naive and semi-immune groups consisted of 11 Exp + 5 Ctrl (Fig. 1). The median age was 30 years for women (19–44 range) and 32 years for men (20–43 range).

**Vaccine and CHMI safety**. The vaccine was safe and well-tolerated. Local pain was the most frequently (75%) reported adverse event (AE) during vaccination (24/32), followed by headache in 31.25% (10/32) and malaise in 31.25% (10/32), which resolved in all cases the next day. These AEs were scored as mild (grade 1) to moderate (grade 2) according to the FDA guidelines[28]. Pain occurred more frequently in the Exp (16 subjects) than in the Ctrl (8 subjects) groups. Fever, nausea, chills, diarrhea, and abdominal pain occurred at low frequencies during the vaccination period (Table 2).

Mild to moderate biochemical or hematological laboratory-related AE were observed. Mild anemia (10.7–11.5 g/dL) occurred in two naive females and two semi-immune (one male/one female) volunteers after the first immunization (normal values >12 g/dL); however, all volunteers normalized before the CHMI. Transient low-level proteinuria (Grade 1–2) was observed after the second immunization in the naive group but reached normal values in the following month. Two semi-immune volunteers presented prolonged thromboplastin time (37.7 and 39.3 s; normal value: 25–35 s). After the third immunization, a volunteer showed glycosuria of 500 mg/dL and glycosylated hemoglobin of 8.1% (HbA1c: normal value: 5.7%) and was diagnosed with diabetes mellitus unrelated to vaccination (Supplement Note 2). All laboratory tests for the remaining participants continued within the normal range. The CHMI was well-tolerated, with no related SAEs recorded.

**Vaccine immunogenicity**. Positive ELISA using the N- and C-peptides indicated seroconversion of all 22 naive and semi-immune Exp volunteers after the first immunization. However, despite their previous parasite exposure, lower reactivity was observed in the semi-immune group than in the naive group. The second immunization induced a moderate boosting of antibodies to the amino (N-) and carboxyl (C-) fragments, as well as priming of the response to the R region. After the third immunization, a slight but significant boosting of antibodies to the three fragments

**Table 1 Baseline characteristics of the vaccine groups and volunteers.**

| Phase | Group | Gender n | Ethnic group n | Age range |
|---|---|---|---|---|
| IIa | NAIVE | F = 7 M = 4 | Mestizo = 11 | 20–43 |
| | CONTROL | F = 3 M = 3 | Mestizo = 6 | 21–42 |
| IIb | SEMI IMMUNE | F = 6 M = 6 | Mestizo = 8 Afro-Col = 3 Indigenous = 1 | 20–44 |
| | CONTROL | F = 4 M = 2 | Mestizo = 5 Indigenous = 1 | 19–28 |

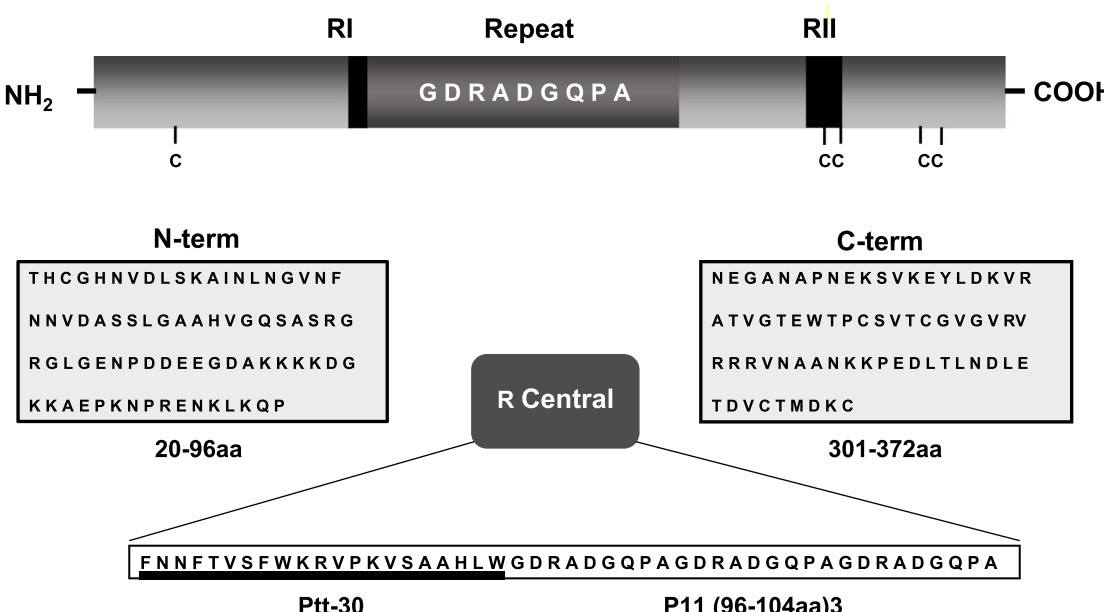

**Fig. 2 Amino-acid sequences of the N, R, and C fragments.** Sequence and localization of the three *P. vivax* CS LSP (N, R, and C) fragments used as immunogens. The N polypeptide corresponded to N-terminal amino acids (aa) 20–96 (N-term), and the C peptide to C-terminal aa 301–372 (C-term). The R peptide VK210 (type I) corresponded to a construct based on the first central repeat (aa 96–104) in tandem three times and collinearly linked to a universal T-cell epitope (ptt-30) derived from tetanus toxin. For the first dose a peptide mixture of N-terminal (term) and fragments (1N:1C) (50 µg/each peptide) was used, whereas for the second and third doses the peptide mixture included N-term, C-term, and R fragments (1N:1C:1R) (50 µg/each peptide).

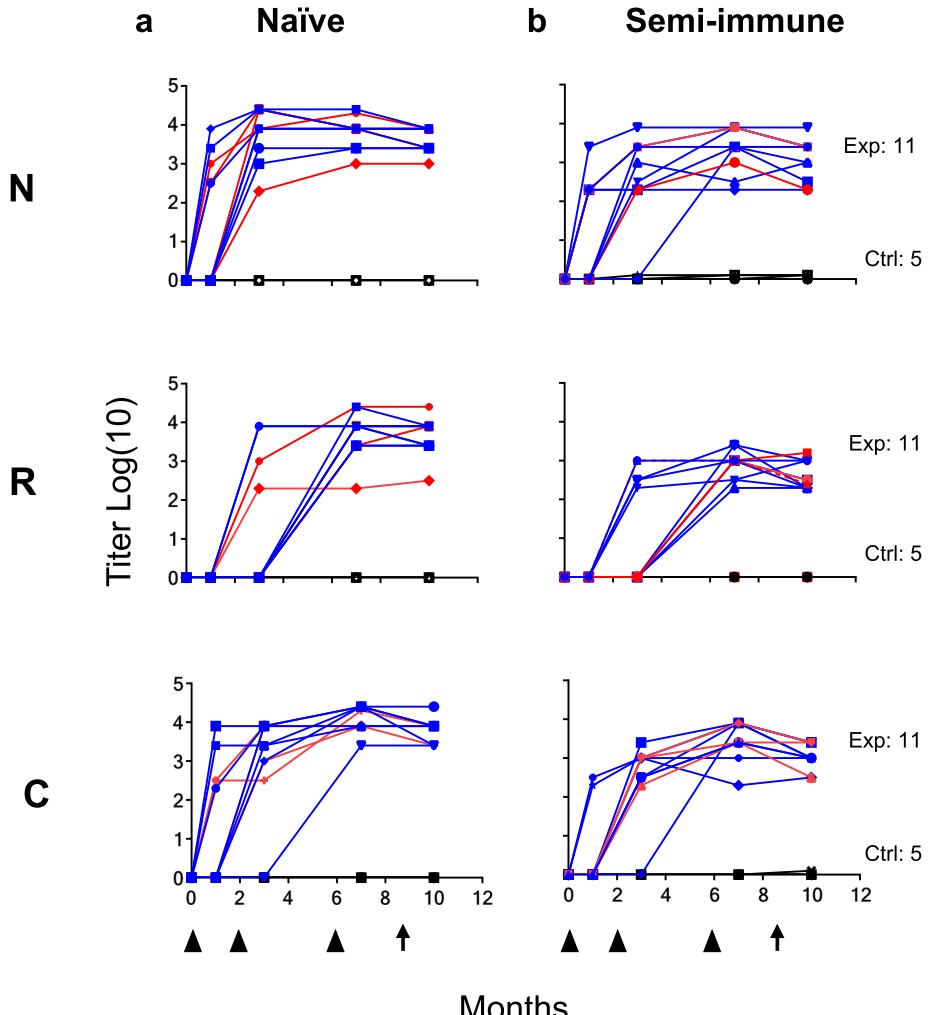

**Fig. 3 Anti-*Pv*CS LSP antibody response in naive and semi-immune volunteers.** Kinetics of specific IgG response to N, R, and C fragments in both Exp ($n = 11$) and Ctrl ($n = 5$) volunteers in Naive (**a**) and Semi-immune groups (**b**). Sterilely protected volunteers are shown in red lines and non-protected in blue lines. Symbols indicate IgG titer as log10 of ELISA values throughout the 10 months of the study. Black triangles at the bottom of the figure indicate immunizations (at 0, 2, and 6 months) and black arrows the controlled human malaria infection (CHMI, at month 9). Significant boosting of antibodies to the three fragments was higher in the naive than in the semi-immune groups (N-, $p = 0.046$; R-, $p = 0.0013$; C-, $p = 0.00505$). Antibody titers did not associate with infection intensity in the naive group (N, $p = 0.72$; C, $p = 0.55$; R, $p = 0.65$) or semi-immune group (N, $p = 0.98$; C, $p = 0.73$; R, $p = 0.52$). Source data are provided as a Source data file.

**Table 2 Number of volunteers reporting vaccine-related adverse events in experimental and control groups.**

| Adverse events[a] | Group | Phase IIa | | | | | | Phase IIb | | | | | |
|---|---|---|---|---|---|---|---|---|---|---|---|---|---|
| | | Naive ($n = 11$) | | | Control ($n = 5$) | | | Semi-immune ($n = 11$) | | | Control ($n = 5$) | | |
| | Doses | 1 | 2 | 3 | 1 | 2 | 3 | 1 | 2 | 3 | 1 | 2 | 3 |
| *Local* | | | | | | | | | | | | | |
| Injection site pain | | 5 | 4 | 5 | 3 | 2 | 3 | 1 | 3 | 3 | 1 | 2 | |
| Swelling | | 1 | 1 | | | | | | | | | | |
| *Systemic* | | | | | | | | | | | | | |
| Headache | | 2 | | 1 | 3 | 1 | 1 | 1 | 1 | 1 | 1 | | |
| Malaise | | 3 | 1 | | 2 | | 1 | | | | | 2 | 1 |
| Fever | | 2 | | | 2 | | | 1 | | | | 1 | |
| Nausea/Emesis | | 2 | | | 1 | | | | 2 | | | | |
| Chills | | | | | | | | | | 2 | | | |
| Diarrhea | | 3 | | | 1 | | | 1 | | | | | |
| Abdominal pain | | 1 | | | 1 | | | | | | | | |

The number of individuals in phases IIa and IIb study groups who developed local and systemic Adverse Events (AE) in Exp ($n = 11$) and Ctrl ($n = 5$) volunteers is indicated. Transient (1 day) pain at injection site (75%) and headache and malaise (31.25%) were observed with mild (grade 1) to moderate (grade 2) intensity.
[a]All AE were graded I–II.

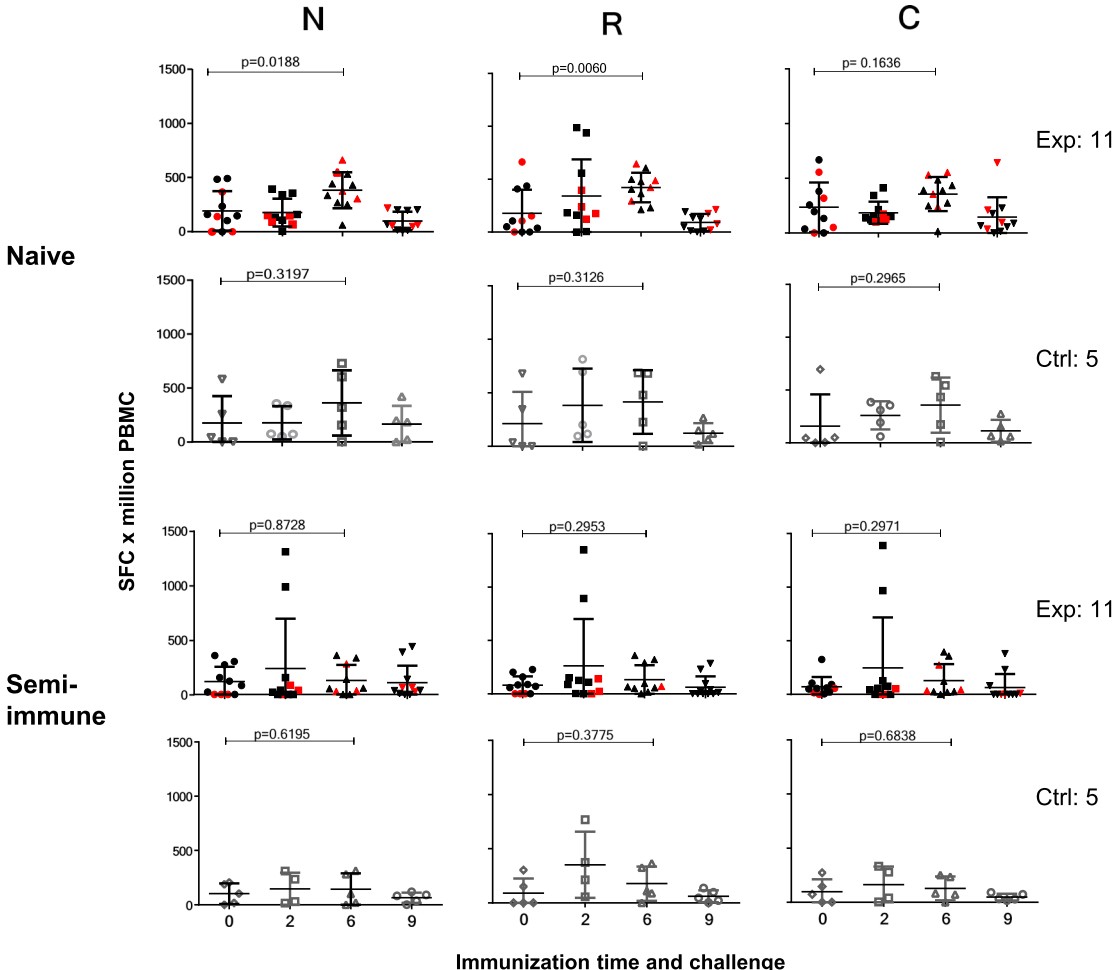

**Fig. 4 Specific induction of IFN-γ by *Pv*CS-LSP in naive and semi-immune groups.** Single-cell interferon gamma (IFN-γ) ex vivo production by fresh peripheral blood mononuclear cells (PBMC) from naive ($n = 17$) and semi-immune ($n = 18$) volunteers collected before immunization (0, 2, 6 months) and CHMI (month 9). Values are expressed as mean of IFN-γ (spot-forming cells) sfc/log10$^6$ in response to 40 h of in vitro stimulation with each *Pv*CS protein fragment (N, R, C). Cells produced IFN-γ upon stimulation with the different fragments in an unstable manner throughout the study phases but with significant differences between naive and semi-immune volunteers (N, $p = 0.046$; R, $p = 0.0013$; C, $p = 0.005$. Only peptides N ($p = 0.0188$) and R ($p = 0.0060$) displayed significant differentes between first and third immunization. Neither naive (N, $p = 0.67$; C, $p = 0.94$; R, $p = 0.15$) nor the semi-immune (N, $p = 0.76$; C, $p = 0.71$; R, $p = 0.48$) IFN-γ levels were associated with parasitemia. Red symbols denote sterilely protected volunteers. Source data are provided as a Source data file.

was observed in some volunteers, higher in the naive than in the semi-immune groups (N-, $p = 0.046$; R-, $p = 0.0013$; C-, $p = 0.00505$); all control volunteers remained seronegative during the immunization and infection phases (Fig. 3). Previous malaria experience of the semi-immune volunteers with *P. vivax* was initially confirmed by their immunofluorescence (IFAT) and enzyme-linked immunosorbent assay (ELISA) response to parasite blood-forms antigens, although all volunteers were negative to the *Pv*CS.

A regression analysis of the antibody titers to the protein fragments (N, R, C) determined by ELISA on months 7 and 10 indicated that neither in the naive group (N, $p = 0.72$; C, $p = 0.55$; R, $p = 0.65$) nor in the semi-immune group (N, $p = 0.98$; C, $p = 0.73$; R, $p = 0.52$) titers presented association with infection (parasitemia levels).

Likewise, the single-cell IFN-γ production by PBMC in response to each protein fragment was significantly different between naive and semi-immune volunteers (N, $p = 0.046$; R, $p = 0.0013$; C, $p = 0.0051$). IFN-γ production developed in all volunteers upon cell stimulation with the protein fragments in an unstable manner throughout the immunization phase, indicating

no clear boosting trend through the immunizations period. However, a slight increase of IFN-γ levels developed at time 2 in the naive group against N ($p = 0.0188$) and R ($p = 0.0060$) peptides (Fig. 4). Nevertheless, neither the naive (N, $p = 0.67$; C, $p = 0.94$; R, $p = 0.15$) nor the semi-immune IFN-γ levels (N, $p = 0.76$; C, $p = 0.71$; R, $p = 0.48$) determined at month 7 before CHMI, were associated with parasitemia. Additionally, there was no boosting of the IFN-γ response after sporozoites CHMI; instead, there were decreased cytokine levels in both groups after CHMI. This decrease was highly significant in the naive group (N-peptide, $p = 0.0022$, R-, $p = <0.0003$, and C-, $p = 0.0367$) whereas it was non-significant in the semi-immune group (N-peptide, $p = 0.902$, R- $p = 0.263$, and C-, $p = 0.371$) (Fig. 4). Overall, no association was observed between antibody titers and IFN-γ together and parasitemia.

**Vaccine efficacy.** Regarding the primary outcome corresponding to the vaccine's protective efficacy, clinical manifestations consistent with malaria such as fever, chills, headache, and profuse sweating were shown from days 14 to 19 after CHMI by Ctrl (5/5)

**Table 3 Description of the CHMI and infection outcomes.**

| Phase | Group | Code | Doses | Mosquitoes | | Prepatent period (days) | Parasite/µL (microscopy) |
|---|---|---|---|---|---|---|---|
| | | | | Bites[b] | Spz[c] | | |
| IIa | NAIVE | CS1001 | 3 | 4 | 32 | 14 | 120 |
| | | CS1006 | 3 | 2 | 32 | 16 | 100 |
| | | CS1013 | 3 | 2 | 32 | 17 | 100 |
| | | CS1015 | 3 | 2 | 32 | 15 | 100 |
| | | CS1023 | 3 | 3 | 22 | 16 | 80 |
| | | CS1025 | 3 | 2 | 25 | 16 | 60 |
| | | CS1028 | 3 | 3 | 22 | 16 | 280 |
| | | CS1030 | 3 | 3 | 46 | P | 0 |
| | | CS1031 | 3 | 2 | 10 | P | 0 |
| | | CS1036 | 3 | 2 | 100 | P | 0 |
| | | CS1038 | 3 | 2 | 100 | P | 0 |
| | CONTROL | CS1003 | 3 | 3 | 215 | 16 | 400 |
| | | CS1005 | 3 | 2 | 32 | 16 | 100 |
| | | CS1012 | 3 | 3 | 100 | 15 | 220 |
| | | CS1018 | 3 | 3 | 32 | 16 | 240 |
| | | CS1037 | 3 | 3 | 22 | 16 | 100 |
| | | CS1016 | 2[a] | NA | NA | NA | NA |
| IIb | SEMI-IMMUNE | CS1506 | 3 | 2 | 100 | 12 | 60 |
| | | CS1511 | 3 | 2 | 32 | P | 0 |
| | | CS1535 | 3 | 2 | 100 | P | 0 |
| | | CS1537 | 3 | 2 | 316 | 19 | 20 |
| | | CS1538 | 3 | 2 | 1000 | 15 | 60 |
| | | CS1547 | 3 | 2 | 32 | P | 0 |
| | | CS1553 | 3 | 3 | 22 | 19 | 400 |
| | | CS1565 | 3 | 3 | 100 | 15 | 160 |
| | | CS1569 | 3 | 2 | 100 | 16 | 340 |
| | | CS1575 | 3 | 2 | 100 | 14 | 128 |
| | | CS1581 | 3 | 2 | 10,000 | 17 | 60 |
| | | CS1584 | 1[a] | 2 | NA | NA | NA |
| | CONTROL | CS1549 | 3 | 4 | 56 | 17 | 50 |
| | | CS1554 | 3 | 2 | 100 | 16 | 400 |
| | | CS1570 | 3 | 3 | 22 | 17 | 70 |
| | | CS1572 | 3 | 2 | 1000 | P | 0 |
| | | CS1574 | 3 | 2 | 32 | 17 | 220 |
| | | CS1579 | 1[a] | 2 | NA | NA | NA |

*P* protected, *NA* not apply.
[a]Withdraws.
[b]No. of infected mosquitoes.
[c]Spz density.

and Exp (7/11) individuals in the naive group (phase IIa) and from days 12 to 19 by Ctrl (4/5), and Exp (8/11) in semi-immune volunteers (phase IIb), which lasted until malaria treatment. The naive Exp group ($n = 11$) was composed of 36.36% men and 63.64% women, while the Ctrl group ($n = 5$) had 20% men and 80% women, aged 19–43 years. In these groups, patent parasitemia developed between days 14 and 17 (mean 15.7 days), with no difference between the Ctrl (15.8) and Exp (15.7) volunteers. Furthermore, all naive-Ctrl volunteers (5/5) (100%) were infected, whereas, in the Exp group, a total of 6/11 (54.5%, 95% CI 0.25–0.84) individuals displayed a reduction of the parasite load <100 parasites/µL. This parasitemia level was considered the threshold as it was below the minimal parasitemia in naive Ctrl (100 parasites/µL) and corresponded to quartile 1 (25% of total data). Notably, 4/6 (66%, 95% CI 0.38–0.94) of the protected volunteers did not develop patent parasitemia over the 60 days follow-up, indicating overall sterile protection of 4/11 (36%, 95% CI 0.08–0.64) in the phase IIa trial; and general vaccine efficacy of 55% (1- RR = 0.45) (Table 3 and Fig. 5).

In the semi-immune Exp group ($n = 11$), 45.45% were men and 54.55% women, while the Ctrl group ($n = 5$) had 20% men and 80% women, all volunteers aged between 19 and 28 years. In this phase, patent parasitemia developed between days 12 and 19

(mean 16.1 days), with no difference between the mean patency of the semi-immune Ctrl (16.7) and Exp (17.6) volunteers. Using the protection cutoff of the Phase IIa study, three volunteers of the semi-immune Ctrl group (total 3/5) (60%, 95% CI 0.31–0.89) were considered protected as parasitemia levels were ≤100 parasites/µL, including the volunteer (CSI 572) that did not develop patent parasitemia. Likewise, 3/7 semi-immune Exp volunteers (43%, 95% CI 0.14–0.72) did not develop patent parasitemia over the 60 days follow-up. The reduction in infection intensity and frequency in the Ctrl (60%, 95% CI 0.31–0.89) and Exp (64%, 95% CI 0.35–0.92) groups indicated no difference. However, when the prepatent periods of the Exp naive (15.8) and semi-immune (17.6) groups were compared, we found that there was a significant difference ($p = 0.0034$) (Table 3 and Fig. 5). Unexpectedly, one of the semi-immune volunteers of the Ctrl group (CS1506) developed parasitemia about 2 months after returning to the endemic area; we could not determine whether it was due to reinfection or relapse.

## Discussion

This study confirmed the safety, tolerability, and immunogenicity observed in the previous phase I vaccine studies with similar *P. vivax* CS-derived LSP formulations[18,19]. More importantly, it

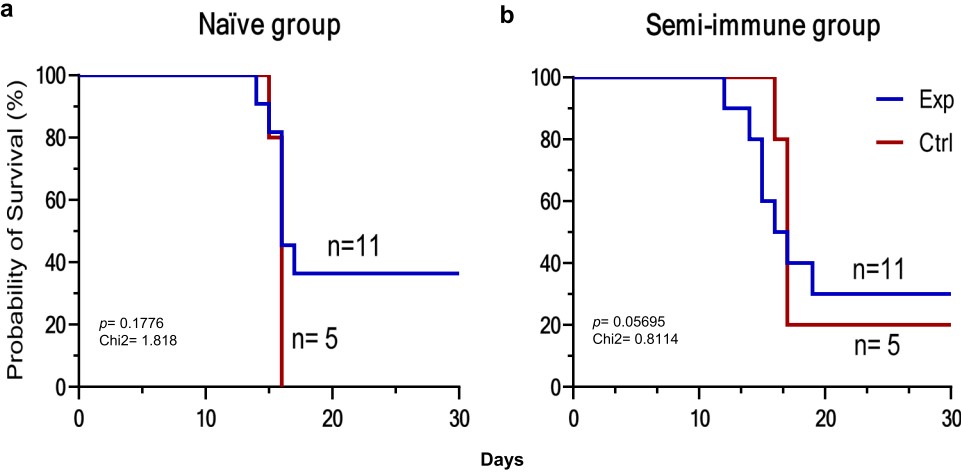

**Fig. 5 Survival curve for naive and semi-immune groups.** Protective efficacy Kaplan–Meier curves are shown for naive (**a**) and semi-immune (**b**) groups. One volunteer of the semi-immune Ctrl group (red line) did not develop parasitemia. Exp: Experimental (blue lines), Ctrl: Control. Prepatent periods of the Exp naive (15.8) and semi-immune (17.6) groups presented significant differences ($p = 0.0034$). Source data are provided as a Source data file.

demonstrated a protective efficacy of 54.5% in malaria-naive volunteers, 36% of whom displayed sterile immunity (phase IIa). Furthermore, the phase IIb component analysis indicated an even greater reduction of parasitemia frequency and intensity in the Exp group (64%, 95% CI 0.35–0.92); however, the reduction observed in 3/5 volunteers of the corresponding Ctrl group (60%, 95% CI 0.31–0.89), makes it difficult to establish protection in this phase independently. Nevertheless, when compared with the phase IIa (naive) groups, the Phase IIb group (semi-immune) displayed a 2 days delay (17.6) ($p = 0.0034$) in the prepatent period, which is similar to that induced by the recombinant VMP001 *Pv*CS vaccine formulation in 59% of the naive vaccinees[29]. In addition, the number of volunteers with lower parasitemia and sterile immunity indicate a significant protective efficacy of the *Pv*CS, both in naive (Phase IIa) and semi-immune volunteers (Phase IIb).

Although the outcome of the phase IIb component is less conclusive, the result is encouraging and has provided valuable bases for further Phase IIb trials with larger groups, subjected to CHMI, directly in endemic areas.

Importantly, AEs related to vaccination occurred with similar frequency in all groups. AEs were limited to local, transient pain at the vaccine injection site, similar to those reported before with commercially available vaccines[30] recombinant *Pv*CS VMP001[26] and phase I *Pv*CS LSP formulated Montanide ISA-51[14,31]. This tolerability is in contrast with unacceptable reactogenicity previously described in HIV- and malaria *Pfs*-25 vaccines formulated in Montanide adjuvants[32]. Regarding the infection experienced by one of the semi-immune Ctrl participants after returning to the endemic area, it may have corresponded to a reinfection. Unfortunately, the volunteer was in a rural setting, and the parasites were not available for genotyping; this patient was one of the not protected volunteers.

In terms of immunogenicity, these studies confirmed the results of previous phase I trials where all volunteers displayed the induction of antibody and IFN-γ specific responses to the three LSP in a fragment-balanced fashion; both the antibody patterns and the unstable behavior of IFN-γ resembled those of the previous trials[18,19]. The lower antibody levels of the semi-immune groups (N-peptide, $p < 0.045$; R-, $p < 0.01$; C-, $p < 0.016$) as well as the IFN-γ had also been reported[20,23]. Although the antibody response was higher in the naive group, we did not find any association between antibody titers and specificity to the N, R, or C at the time of the CHMI, and protection. The lack of

significance between protection and immune response may be due to the small size of the final groups, as well as to the highly variable immune response among participants, likely due to the population heterogeneity in terms of ethnicity, sex, and nutrition status among other potential factors. Moreover, the results are in agreement with a previous study[26] where no correlation was found when we analyzed antibody titers and IFN-γ responses but continues to be intriguing as both effector mechanisms have been experimentally shown to be associated with protection against *Plasmodium* pre-erythrocytic infection[33,34].

The hypo-responsiveness of the semi-immune Exp groups in terms of specific IgG and IFN-γ to the immunogens is also in contrast with the delay in patency displayed by this group of volunteers. However, the identification of surrogate markers or signatures of immune protection remains elusive even by using systems biology[35].

The lack of immune boosting by CHMI is likely due to the brief exposure to a presumably low number of sporozoites (Table 3). Surprisingly, the significant IFN-γ decrease in the naive group (N-peptide, $p = 0.0001$, R-, $p = <0.0001$, and C-, $p = 0.0367$) (Fig. 4) might be related to the volunteers' exposure to mosquito's saliva. Some studies have shown that after biting, saliva can trigger different effects on humans' immune cells in mice grafted with human hematopoietic stem cells directly by the Th1/Th2 response[36,37].

The contrast between the outcomes of phase IIa and IIb appears to be related to (1) the reduced number of volunteers remaining in the Ctrl group ($n = 4$), which reduced the power of the study, (2) the volunteers' ethnic heterogeneity in both Ctrl and Exp groups, (3) a possible dysregulation of the semi-immune volunteers; and potentially other factors[35], generating a great heterogeneity, and (4) the semi-immune status of phase IIb volunteers, where protection mechanisms may be different from the ones studied here, which warrants further studies with larger groups.

Two main features make the design of these *P. vivax* clinical trials different from those previously reported with *P. falciparum*[11] and *P. vivax* vaccine candidates[19,29], (1) both phase IIa and phase IIb were simultaneously conducted, and (2) we deliberately avoided inclusion of the *Pv*CS R-fragment in the first vaccine dose to diminish its immunodominance. The rationale behind simultaneously conducting phase IIa/phase IIb trial is based on Cali being a malaria-free city in Colombia with conditions to explore the feasibility of comparative vaccine trials under

similar conditions, and Buenaventura a malaria-endemic town, located at ~70 miles from Cali. In addition, the response of naive and semi-immune volunteers to *P. vivax* sporozoite CHMI had been already evaluated[23]. Furthermore, the geographic and epidemiological characteristics provided privileged conditions for this proof-of-principle on the feasibility of closely controlled phase IIb *P. vivax* pre-erythrocytic vaccine trials.

The overall data are valuable in different ways, (1) they indicate an important degree of protection (54.5%) in the naive vaccinated group, with remarkable sterile immunity (36%), and potentially greater efficacy (≥60%) in semi-immune populations, (2) they highlight the importance of CHMI to ensure volunteers' exposure to parasites in a controlled manner, (3) they provide a foundation to further cost-effective phase II trials, and (4) they warrant efficient comparison of the immune response, i.e., using systems biology and system-level data analysis for comparing populations exposed to different environmental conditions[35].

Recent studies on the response of individuals from malaria-endemic and non-endemic areas to *Pf*-RTS,S[38,39] showed a similar hypo-responsiveness, suggesting that individuals from malaria-endemic regions, either actively infected or not, display an altered basal immune status with a paucity of regulatory mechanisms and altered memory cell function leading to lower responsiveness to vaccines[35,37,40,41]. It has been hypothesized that this corresponds to an immunological imbalance caused by permanent exposure to malaria parasites, mosquito bites, and other host and environmental factors such as sex, ethnicity, nutrition, malaria endemicity, and others that may influence the host's immune response and immunity to malaria in endemic areas. Although these studies do not provide further light on the correlates of protection, they generated reagents for further identifying or confirming potential critical *Pv*CS epitopes.

One of the limitations of this study was volunteer retention. Although semi-immune volunteers were requested to be outside the endemic area for the parasitemia follow-up, some decided to return to their homes about a month after CHMI, making follow-up more challenging. Nevertheless, it provided a unique opportunity to directly assess the vaccine candidate's protective efficacy in volunteers previously exposed to natural *P. vivax* infections, and generated preliminary data for further and larger Phase IIb trials.

The differential protective efficacy achieved here may have been influenced by ethnic factors and the final limited sample size. While all naive Exp and Ctrl volunteers were mestizo, the semi-immune group was heterogeneous (mestizo, afro-descendent, indigenous). Alternatively, one could hypothesize that the levels of premunition to *P. vivax* malaria in endemic areas contributed to the vaccine-elicited immunity and therefore it has to be evaluated differently. Because the study sample was not representative of the whole population of individuals living in the country's malaria-endemic areas, a comparative trial with larger volunteer groups is being prepared (NCT 04739917) in malaria-endemic and non-endemic regions and will address ethnicity; that trial will include more powerful system biology and system data analyses. A highly efficacious *P. vivax* pre-erythrocytic vaccine is of utmost importance due to the high relapsing rate of this parasite species in some regions of the world[7].

## Methods

**Ethics statement.** The study protocol was reviewed and approved by the Institutional Review Boards of the Malaria Vaccine and Drug Development Center (MVDC-CECIV) and Centro Médico Imbanaco (CMI # 0992304-493-26202) (Supplement Note 1). The study complied with the Declaration of Helsinki principles, International Conference on Harmonization, Good Clinical Practices guidelines, and all pertinent Colombian regulations. All participants provided written informed consent (IC) and were advised that they were free to withdraw from the study at any time. Volunteers were excluded if they had diseases or

medical conditions that would alter the vaccine's assessment or any condition that could increase the risk of adverse outcomes.

**Subjects selection and enrollment.** All study participants complied with the following general criteria. (1) Healthy men and women aged 18–45 years; (2) participants freely and voluntarily providing signed IC, accompanied by two witnesses signatures; (3) for women, adequate contraception from the time of enrollment; (4) accept not to travel to areas considered endemic for malaria during the infection follow-up period (1 month); (5) be reachable by phone throughout the study period; (6) being Duffy blood group positive (Fy+); (7) harboring Hb levels >11 g/dL; (8) availability to attend all visits during the study period; (9) not participating in other clinical studies. Naive volunteers had no history of malaria infection, whereas semi-immune volunteers had a history of previous malaria infection(s) and positive serological tests for *P. vivax* malaria. Volunteers were excluded from enrollment as naive if they had a history of having lived in a malaria-endemic area for the past 6 months, and excluded from the semi-immune group if they had negative IFAT (<1:20) for *P. vivax*.

**Parasite donors.** Malaria patients accepting to provide *P. vivax* infected blood for sporozoite production had to comply with the following criteria: (1) Be healthy men and women aged 18–45 years; (2) harbor *P. vivax* infection with parasitemia ≥0.1% as determined by thick blood smear examination; (3) not having *P. falciparum* or *P. malariae* circulating malaria parasites; (4) Hb levels ≥9 g/dL at the time of malaria diagnosis; (5) have the capacity to provide IC freely and voluntarily. Should he/she be illiterate, accept to assert the decision to participate by a fingerprint in IC form; (6) if the potential donors were minors aged 15 to 17 years, sign the IC, and one of his/her parents must sign the IC, accompanied by two additional witness signatures. Potential parasite donors were excluded from enrolment if (1) they had negative IFAT (<1:20) for *P. vivax* in screening tests, (2) if patients had chronic or acute disease, different from malaria by *P. vivax;* (3) hemoglobin levels <9 g/dL at the time of recruitment, (4) having received anti-malarial treatment before blood draw; (5) having a history of disease or clinical conditions that according to medical criteria might significantly increase their risk by participating in the study.

**Study design and participants.** This was a comparative phase IIa/IIb randomized, double-blind, controlled trial conducted in Colombia to evaluate the safety and protective efficacy of *Pv*CS formulated in Montanide ISA-51. Thirty-five healthy, Fy+ men and non-pregnant women (19–44 years of age) were recruited from a larger group ($n = 121$) and allocated into two groups: phase IIa with healthy malaria-naive ($n = 17$), all mestizo (seven male) with age range of 20–43 years; and phase IIb with malaria semi-immune ($n = 18$) divided into afrodescendants ($n = 3$), mestizo ($n = 13$), and indigenous ($n = 2$), with age range of 19–44 years of age. Volunteers were recruited as before[23,27] using several communication strategies, including mass media approaches. Participants were recruited from October 3, 2014 (first patient) to December 22, 2014 (last patient) and were randomly (simple) assigned in a 2:1 ratio based on the pre-specified (study protocol- Supplement Note 1) inclusion and exclusion criteria. A blinded data manager controlled the allocation to receive the vaccine (Experimental; Exp, $n = 25$) or placebo (Control; Ctrl, $n = 10$) (Fig. 1). Access to the randomization code was strictly controlled at the pharmacy.

The naive group ($n = 17$) was further divided into Exp ($n = 12$) and Ctrl ($n = 5$), as was the semi-immune group [$n = 18$ into Exp ($n = 13$) and Ctrl ($n = 5$)]. Naive volunteers were from Cali (Capital of Valle del Cauca department), a malaria-free area, located at the southwest of Colombia at ~1.000 m.a.s.l. Volunteers were eligible based on no history of malaria and negative *P. vivax* serology. The semi-immune volunteers were recruited in Buenaventura, the main port on the Pacific coast of Colombia, at 80 km from Cali with an altitude of 7 m.a.s.l. The region is a low to moderate malaria-endemic area were both *P. vivax* and *P. falciparum* parasites are transmitted. Selected semi-immune volunteers had a history of malaria and antibodies against *P. vivax* blood stages with titers by immunofluorescence (IFAT ≥ 1:20) or enzyme-linked immunosorbent assay (ELISA ≥ 1:200) against a *P. vivax* recombinant MSP-1 protein[42].

**Vaccines.** The three LSP (N, R, and C) synthesized under good laboratory practices (GLP) conditions at the Biochemistry Institute, University of Lausanne, Switzerland, were packaged, lyophilized, and tested for sterility and apyrogenicity (Pharmacie, Centre Hospitalier Universitaire Vaudois, Laussane, Switzerland). As previously described[18], the N polypeptide corresponded to N-terminal amino acids (aa) 20–96 (N-term), and the C peptide to C-terminal aa 301–372 (C-term). In contrast, the R peptide VK210 (type I) corresponded to a construct based on the first central repeat (aa 96–104) in tandem three times and collinearly linked to a universal T-cell epitope (ptt-30) derived from tetanus toxin[14,16,43] (Fig. 2). For the first dose, a peptide mixture of N-term and C-term fragments (1N:1C) (50 μg/each peptide) was used, whereas for the second and third doses, the peptide mixture included N-term, C-term, and R fragments (1N:1C:1R) (50 μg/each peptide). In this immunization schedule, we deliberately wanted to diminish the levels of antibodies to the R region known to be immunodominant[44], to induce a balanced response to the three protein fragments[19]. According to manufacturer

recommendations on the same day of subject immunizations, peptide mixtures were emulsified in Montanide ISA-51 (VG, code 36362Z, Seppic, Paris, France) in the same proportion according to manufacturer recommendations on the same day of subject immunizations. Saline solution (Baxter, Deerfield, IL, USA) was emulsified with the same adjuvant and used as a placebo.

**Interventions**. The primary outcome was the *P. vivax* CS LSP vaccine's protective efficacy against the *P. vivax* CHMI in malaria-naive and semi-immune volunteers as determined by the parasite reduction in terms of frequency and density, and the secondary outcome was the B and T-cell immune response associated with protection. During the immunization period, blood samples were collected before the first immunization and at several time points thereafter to study the immune response and the blood chemistry. Enrolled participants received three vaccine doses (Exp group) or placebo (Ctrl group) at months 0, 2, and 6 by i.m. injection in the deltoid muscle with a volume of 0.5 mL. Vaccines were prepared by staff researchers not involved with patient care.

**Safety**. At enrolment, blood samples were collected to determine Fy blood group and G6PD deficiency. Fy+ was confirmed to ensure the volunteer's susceptibility to *P. vivax* blood infection[45], and normal G6PD status to prevent the risk of hemolytic anemia upon *P. vivax* treatment with primaquine[46,47]. Complete blood count (CBC), prothrombin time (PT), partial thromboplastin time (PTT), alanine aminotransferase (ALT), aspartate aminotransferase (AST), bilirubin, alkaline phosphatase, blood urea nitrogen (BUN), creatinine, and pregnancy were performed. During the immunizations period, volunteers were under direct medical supervision during the hour following immunization to detect any adverse reaction to the vaccine injection, after which a physical examination was performed. Eight hours post-injection, volunteers' physical status was assessed by a telephone call. Also, a personal follow-up was conducted 1 week before the following immunizations. Clinical laboratory tests evaluated the vaccine tolerability and safety at months 0, 1, 2, 3, 6, 7, 9, and 10. Volunteers were also under observation for 1 h after the CHMI and then by phone monitoring 8 h after and once a day until day 4[23]. Volunteers were then evaluated daily for clinical manifestations and microscopic patent parasitemia from days 5 to 30 after the challenge and every second day until day 60. Two experienced, independent microscopists evaluated parasitemia by counting the number of asexual *P. vivax* parasites per 400 white blood cells (WBC), assuming normal WBC counts (8000 cell/μL). Samples were considered negative after observing 200 microscopic fields, and qPCR was performed subsequently for retrospective analysis. Adverse events (AE) were recorded, graded, and classified according to FDA recommendations[28].

**Sporozoite production**. Whole blood (15 mL) was collected by venipuncture (Vacutainer tubes, Becton Dickinson, NJ, USA) from patients diagnosed with VK210 *P. vivax* in Leticia, Colombia, and used to infect colonized *Anopheles albimanus* mosquitoes. Fed mosquito batches were dissected and microscopically examined for the presence of oocysts in the midgut (day 7) and sporozoites in salivary glands (day 14) as previously described[17]. For CHMI, only batches with ≥60% sporozoite infection rates were considered acceptable. CHMI was performed 3 months after the last immunization by volunteers' exposure to 2–4 *P. vivax*-infected mosquito bites[21,22]. After biting, individual mosquito dissection confirmed the presence of blood in midguts and sporozoites in salivary glands[22].

**Humoral response**. Antibody response was assessed using blood samples collected on months 0, 1, 3, 7, and 10 before immunization and CHMI and measured by ELISA with N, R, or C peptides (1 μg/mL) antigens[18]. Controls were selected from a pool of sera from semi-immune blood donors (positive ctrl) and a pool of sera from malaria-naive donors (negative ctrl). In addition, parasite recognition by anti-LSP antibodies was determined by IFAT using *P. vivax* fixed sporozoites[18]. Previous malaria contact in the group of semi-immune volunteers was confirmed by IFAT using parasite blood forms and ELISA test using a *Pv*-MSP-1 recombinant protein[42].

**IFN-γ ELIspot production**. Peripheral blood mononuclear cells (PBMC) were separated immediately after whole blood was collected by venipuncture, at months 0, 1, 3, 7, and 10, using Ficoll-Histopaque (Sigma-Aldrich, St. Louis, MO, USA) density gradients. PBMC was used to determine the ex vivo production of IFN-γ, as previously described[19]. Briefly, fresh PBMCs ($4 \times \log 10^5$/well) were mixed with 10 μg/mL of each LSP, and after 40 h culture, the number of IFN-γ spot-forming cells (sfc) was counted using an ELIspot reader (AID Autoimmun Diagnostika GmbH, Germany); results were expressed as the mean number of IFN-γ sfc per $\log 10^6$ PBMC. In each assay, PBMC were also incubated in the absence of peptides to determine the IFN-γ background production level. The background level of each test was subtracted from the specific counts obtained from peptide stimulated cells. Volunteers were considered responders if the number of sfc in their samples increased from their baseline level; any increase ≥5 sfc was considered positive[19].

**Statistical analysis**. Data were collected and managed using REDCap 5.10.1 (Nashville, TN, USA) electronic data capture tools, analyzed using SPSS version

16.0 software (SPSS Inc., Chicago, IL, USA), and plotted using Graph Pad Prism version 6.0 (GraphPad Software, San Diego, California, USA). The primary outcome evaluated the frequency and density of *P. vivax* infection in volunteers vaccinated with *Pv*CS LSP formulated in Montanide ISA-51. The study sample size was calculated with a confidence level of 95%, taking into account a previous census of ~5603 subjects with recent malaria infections in Buenaventura[48], where the estimated prevalence of the Fy+ genotype (30%) and G6PD deficiency (12%) in the target population indicated that only 1479 subjects would be suitable for participation. Likewise, we estimated that only 3–10% of the subjects (44–147) would volunteer for the trial based on previous studies in the same area.

For the secondary outcome, antibody titers and IFN-γ elicited by immunization were compared among groups at several time points in the study using regression analysis with *p* values ≤0.05 considered significant. A T-student test was used to compare differences between the response to N, R, C fragments in naive and semi-immune groups. Regression analyses were performed both as separate models per peptide (N, R, and C) as well as with grouped (N + R + C) fragments. In addition, Fisher's exact test was used to compare proportions of protective efficacy between Exp and Ctrl in naive and semi-immune groups. The association between immune response and protection was assessed using paired sample T-test. A volunteer was considered protected if the parasitemia reached levels significantly lower than that of the control groups. Parasitemia was treated as a continuous variable.

**Reporting summary**. Further information on research design is available in the Nature Research Reporting Summary linked to this article.

## Data availability

The authors confirm that the data supporting the findings of this study are available within the article and its Supplementary Material. Raw data that support the findings are in the REDCap database, available from the corresponding author upon reasonable request; S.H. (sherrera@inmuno.org). The timeframe for responding to the requests will be ~25 business days. The data are not publicly available because they contain information compromising research participants' privacy/consent. This trial is registered on ClinicalTrials.gov under the identifier NCT02083068. Source data are provided with this paper.

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

## Acknowledgements

We thank the volunteers from Cali and Buenaventura for their invaluable collaboration in the study, as well as the parasites donors from Leticia (Colombia), the continuous support from Dr. Armando Gonzalez-Materon, C.E.O. of the CMI, and the technical support from Nora Cespedes and Andres Vallejo, John García, Lucia Buritica, Eydy Zuñiga, Mary Lopez-Perez, and Laureano Mestra. To Seppic Inc., Paris, France for kindly providing the adjuvant. We also acknowledge Alberto Alzate, Pablo Chaparro, and Brayan Osorio for critical discussion of the manuscript. This trial was funded by Colciencias (grant 529-2009 to S.H.), NHLBI (grant RHL086488 A to S.H.), and the MVDC/CIV Foundation (grant 2014-1206 to M.A.H.).

## Author contributions

Conceptualization: M.A.H. and S.H. Formal analysis: M.A.H., G.C., S.M.H., A.C., and X.G. Investigation: M.A.H., S.H., G.C., S.M.H., M.L.D., M.A.C., J.H.G., A.C., and X.G. Methodology: M.A.H., S.H., G.C., S.M.H., X.G., A.C., J.H.G., J.O., J.C.D., and A.P. Project administration: M.A.H. and S.H. Resources: G.C., M.A.H., and S.H. Supervision: M.A.H., S.H., and J.O. Validation: M.A.H., SH., and J.O. Visualization: M.A.H., S.H., G.C., and S.M.H. Writing: M.A.H., S.H., G.C., and S.M.H.

## Competing interests

The authors declare no competing interests.
