## [Peer Review File · Nature Communications]

Randomized clinical trial to assess the protective efficacy of a Plasmodium vivax CS synthetic vaccineReviewers' Comments:

Reviewer #1:

Remarks to the Author:

The manuscript presents the first results of a *P. vivax* vaccine that demonstrates protective efficacy against infection following controlled human malaria infection challenge.

The following comments are intended to provide clarity to readers concerning study design and to provide rationale for several discrepancies contained in the document.

1. The first immunization dose contained a mixture of peptides N and C only (50µg/peptide; total dose 100µg/dose), whereas for immunizations second and third, the doses comprised peptides N, R and C (50µg peptide/dose; a total of 150µg protein/dose). What is the rationale for this study design? The central repeat region has been associated with protective efficacy against other Plasmodium species including *P. falciparum*, *P. berghei*, *P. yoelli*. It would appear that the authors deliberately wanted to minimize the elicitation of high levels of antibodies against the repeat region.
2. There is confusion on page 13 regarding protective efficacy determined by the presence of patent microscopic parasitemia that was observed in five of 11 naïve Exp volunteers (42%) and four of 11 semi immune volunteers (36%), who did not develop parasitemia during the 60-day follow-up. This does not seem to be consistent with results in Table 2 which showed 4/11 subjects protected in the naïve group versus 3/11 subjects with no parasitemia in the semi-immune group. Please provide clarity on the presentation of these results.
3. An expanded discussion is warranted about the results on reasons that lower IFN-g reactivity was observed in the semi-immune than in the naïve group.
4. There was little to no relationship between protection and levels of antibody titer by subject. As immune correlates of protection is a "hot" topic in vaccine development, whether an association between N or C terminal domains or the central repeat region correlated with risk/protection following CHMI.
5. The Methods section would benefit from providing sequences of the peptides used for immunization (L, repeat,C).
6. Figure 2 should have log rank P value shown
7. Table 2 showed that 2 control subjects withdrew. Were these before or after CHMI?
8. Table 1 should show AEs by proportion since there are different numbers of subjects receiving 1,2,3 doses of vaccines. There is no reference to grade of severity of AEs which should be included.
9. The authors conclude that one subject with recurrence of *P. vivax* malaria was due to reinfection, however there was no genotyping of these parasites to suggest that a relapse of vivax infection stemming from the CHMI may have been responsible for the 2nd infection.
10. Since subjects received different numbers of infectious mosquito bites, a table should have been included for subjects in the control and vaccine groups documenting exactly how many infectious mosquito bites were used. The force of infection is a determinant of likelihood of infection in CHMI.
11. The discussion would benefit from 1) how antibody or CMI (IFN-g) play a role in protection; 2) what modifications may be necessary to improve vaccine efficacy (modification to vivax sequences or alternative adjuvants?); differences in naïve subjects versus semi-immune subjects related to challenge outcome and immune responses

Reviewer #2:

Remarks to the Author:

These are complex studies and important to conduct. Promising results are shown but there are many questions regarding the trial:

- could not find the registration of this particular trial
- the recruitment started in 2014, when was it completed?
- when were the antibody and T cell assays performed?
- the aim was to recruit 52 subjects but ended up recruiting 35, some explanation would be helpful
- how were the 121 subjects asked to participate, how and where were they asked and how many were from the endemic area and how many from non endemic
- then in the trial flow diagram we see 80 did not meet the inclusion criteria, which again raises the question how was the recruitment organized
- the flow diagram should indicate how many from endemic and non endemic area were excluded
- detail of the recruited subjects should be given in more detail and what is meant by "This study sample is not representative of the whole population of individuals living in the country's malaria-endemic areas;"

Points regarding results:

- figures can be improved considerably
- the antibodies shown are of those that react? the comparisons can be better presented
- would be informative to show ELISPOT results for each individual (dot plots?)

Abstract and Discussion:

- not very precise when it comes to immunological reactivity
- no discussion of the discrepancy between immunogenicity and protection
- deeper discussion of the findings in relation to data from other studies - patterns of parasite development, the immunological data

Reviewer #3:

Remarks to the Author:

The authors present a *P. vivax* CHMI vaccine study demonstrating a level of sterilizing protection in trial participants, this seems of high interest to the study of malaria vaccines. The general message of the paper is concise and clear. My main comments focus on the statistical analyses.

Major comments:

I would recommend the authors have a statistician, preferably a co-author, review this manuscript to ensure that the analysis of the results is appropriate. I do not believe there are any irreparable deficiencies in the data and subsequent results, but I highlight several concerns in the statistical analysis below.

1) Issues related to protection efficacy (PE) statistics and subsequent reporting.

- The PE for the semi-immune group needs to account for the infection rate in their control group (See <https://doi.org/10.1093/infdis/jiaa421> as an example). The protected semi-immune control participant cannot simply be excluded from the PE estimate as an outlier.
- There is no formal statistical analysis of PE. At the very least, what is the precision (e.g., 95% CI) of these estimates? (reference above also addresses this)
- There appears to be two different estimates of sterilizing PE in this study. In the abstract, sterilizing PE is reported as 5/11 (note typo in Abstract and Results, should be 45%, not 42%) and 4/11 based on "no parasitemia". This result is repeated in Results section "Vaccine efficacy", but a second set of numbers are also presented "based on the survival analysis": 4/11 and 3/11. The latter is then repeated in the lead Discussion paragraph. Unless there are multiple definitions of protection (e.g.,

sterilizing vs. reduction in parasitemia), there should only be one estimate of PE reported. If there were multiple definitions of protection, they should be more clearly defined and easily referenced.

- Lines 202-206 (Stats Methods), description of sample size calculation is unclear. What endpoint comparison is being powered? What is the range of effect sizes?

- Lines 206: "...protection efficacy was assessed at a 5% significance level and 80% power." Is this related to the sample size calculation? What comparison (ie, effect size) was assessed for 80% power? Why aren't there any formal comparisons of efficacy performed in this manuscript?

2) Comparison of immunological endpoints: it is unclear what statistical comparisons were performed and what criteria were used to determine their inclusion in the Results. All performed tests should be clearly reported so the reader has a scope of the total hypotheses being evaluated. Generally, I was confused whether p-values represented tests over time (times often not clearly stated) or between cohorts; and there were no tables providing this information.

- There are responder definitions for both assays. It is clearly stated that all vaccinated participants were seropositive via ELISA post-all vaccinations. Is this true for the ELISPOT responses? Can a similar a sentence lead those results?

3) In addition to the binary protection-derived endpoints--sterilizing protection and delay in patency--can we learn anything from the qPCR continuous readout? For example, does parasitemia look similar among infected vaccinated and control participants? Correlations between immunological and parasite endpoints are generally of interest, but perhaps that is a follow-up analysis.

- <https://doi.org/10.1371/journal.pcbi.1005255> is a theoretical overview of parasitemia assessment.

Minor comments

- The authors clearly have a great understanding of the evolving vaccine technology, but the introduction would benefit if it had a little more clinical/epidemiology background. How is PE assessed throughout studies, linking PE between field and CHMI studies, etc.. For example, [doi: 10.1016/j.pt.2016.11.001] appears to be a relevant review including description of the authors' past efforts.

- Figure 2: what do the square points denote in the plot?

- Figure 3: what do the shapes denote? Horizontal dashed lines? Would be a nice if the vaccine times were visualized or stated in the caption.

- Figure 4: light spaghetti behind boxes may help highlight the trends. Unclear if CHMI box is all zero or smushed by scaling, log-transforming could help. Consider denoting positive responders.

- Figures 3-4: Helpful to state cutoff value.

Reviewer # 1

Reviewer statement

The manuscript presents the first results of a *P. vivax* vaccine that demonstrates protective efficacy against infection following controlled human malaria infection challenge. The following comments are intended to provide clarity to readers concerning study design and to provide rationale for several discrepancies contained in the document.

QUESTION 1. The first immunization dose contained a mixture of peptides N and C only (50µg/peptide; total dose 100µg/dose), whereas for immunizations second and third, the doses comprised peptides N, R and C (50µg peptide/dose; a total of 150µg protein/dose). What is the rationale for this study design? The central repeat region has been associated with protective efficacy against other Plasmodium species including *P. falciparum*, *P.berghei*, *P.yoelli*. It would appear that the authors deliberately wanted to minimize the elicitation of high levels of antibodies against the repeat region.

RESPONSE 1. *This concept is now included on Vaccines section, page 7, lines 118-120.*

“In this immunization scheduled we deliberately wanted to diminish the levels of antibodies to the R region known to be immunodominant²⁸, to induce a balanced response to the three protein fragments¹⁷.”

*The interpretation of the reviewer is correct. We wanted to balance the levels of antibodies produced against the three protein regions. The central repeats (R) domain of the CS proteins of all studied Plasmodium species is composed of tandem repeats of the same amino-acid sequence [Eg., in *P.vivax* (GDRADGQPA – VK210 variant)¹⁹ or (ANGAGNQPG – VK247 variant)¹⁹]. This structure confers higher immunogenicity to the central R than the amino (N) and the carboxyl (C) regions, generating an imbalance or immunodominance. While the central R domain is associated with protection, the N and C fragments contain important functional stretches such as the RI in the N-terminal region (Coppi, A et al, 2011 PMID:21262960) of the protein and the RII and neighbor sequences in the C-terminal fragment (ref 17), potentially equally crucial for protection.*

In a previous phase I trial using the same immunogens, we had shown that it was possible to balance the recognition of the three protein fragments by using the vaccination regime similar to the one used here (ref 17). We believe that this immunization strategy influences the protective efficacy.

Q 2. There is confusion on page 13 regarding protective efficacy determined by the presence of patent microscopic parasitemia that was observed in five of 11 naïve Exp volunteers (42%) and four of 11 semi-immune volunteers (36%), who did not develop parasitemia during the 60-day follow-up. This does not seem consistent with Table 2, which showed 4/11 subjects protected in the naïve group versus 3/11 subjects with no parasitemia in the semi-immune group. Please provide clarity on the presentation of these results.

R 2. *The reviewer is correct. We have carefully revised the consistency of these data and have introduced changes in table II, page 28, and in the Statistical Analysis section, page 14, lines 276-282. We hope this new text provides more clarity.*

The precise related changes are described below in responses to reviewers 2 and 3.

Q 3. An expanded discussion is warranted about the results on reasons that lower IFN- γ reactivity was observed in the semi-immune than in the naïve group.

R 3. *We have expanded the discussion of this issue in the Discussion section, pages 15 and page 16, (lines 316-319).*

Lower levels of IFN- γ in the semi-immune volunteers were striking. However, there is growing evidence that semi-immune individuals living in malaria-endemic areas display immune dysregulation, resulting in lower immunogenicity of other malaria vaccine candidates, i.e., Pf-RTS, S; Pf-R21 in semi-immune populations than in malaria-naïve individuals from non-endemic areas (ref 38). This phenomenon is currently considered crucial for further malaria vaccine development. Thus, NIAID recently launched an application request (RFA-AI-20-064) to conduct in-depth studies comparing the malaria vaccine-elicited immunity under endemic and non-endemic conditions.

Q 4. There was little to no relationship between protection and levels of antibody titer by subject. As immune correlates of protection is a "hot topic" in vaccine development, whether an association between N or C terminal domains or the central repeat region correlated with risk/protection following CHMI.

R.4: *We have expanded on this issue in the Discussion Section, pages 15 (line 314) to page -16 (line 336) as follows:*

Here, we have found reactivity to the three different protein regions evaluated (N, R, and C). Responses to the three protein fragments were higher in the naïve (phase IIa) groups, although without correlation between antibodies to the individual fragments that could identify a correlate of protection, suggesting that antibodies to multiple critical epitopes are additive and contribute to protection. In our previous study (ref 17), the three protein fragments were shown to induce sporozoite invasion inhibition (ISI assays).

On the other hand, the correlates of protection (CoP) are indeed a "hot topic." The importance of neutralizing antibodies for protection against sporozoite invasion has been demonstrated experimentally in vitro and in vivo (ref 37-38) using monoclonal antibodies. However, protection does not seem to result from a single specificity or cytokine activity either in nature or experimentally in humans. We favor the idea of multiple antibodies to various critical epitopes, in combination with cytokines and potentially other immune effectors. System biology approaches are expected to contribute to this task. Despite recent studies providing valuable data on human naïve and semi-immune volunteers' response to the Pf-RTS,S, and Pf-R21 vaccine candidates, there is no clear understanding of CoP or immune signatures protection

(ref 39- 40). As mentioned above, a recent NIH-RFA precisely addresses this matter, which is common to other vaccines and micro-organisms.

Q.5. The Methods section would benefit from providing sequences of the peptides used for immunization (L, repeat, C).

R.5 *The sequence of the three peptides have been included in Figure 2, page 23*

Q.6. Figure 2 should have log rank P value shown

R.6. *Fig 2 (now Figure 3) was edited*

Q. 7. Table 2 showed that 2 control subjects withdrew. Were these before or after CHMI?

R.7. *Volunteers withdrew before CHMI.*

It was described in the original Figure 1, page 22. It is now also mentioned in Results section, page 11, lines 214-217.

Q. 8. Table 1 should show AEs by proportion since there are different numbers of subjects receiving 1, 2, 3 doses of vaccines. There is no reference to grade of severity of AEs which should be included.

R.8. *The severity of AEs (i.e., I-III) was included in the original table 1 (page 27), also in Results Section (vaccine and CHMI safety), page 11, lines 222-227, and in Discussion Section, page 15, lines 308-311.*

For more clarity, we have changed table 1 on page 27.

All volunteers received the same number of vaccine doses (3).

Q. 9. The authors conclude that one subject with recurrence of *P. vivax* malaria was due to reinfection; however, there was no genotyping of these parasites to suggest that a relapse of *vivax* infection stemming from the CHMI may have been responsible for the 2nd infection.

R.9. *The reviewer is correct. The new case developed two months after the close follow-up had ended. The volunteer was in a rural setting in the endemic area, distant from our main campus in Cali. He was diagnosed and treated at one of the points of care of the National Malaria Control Program. We were informed about it after the patient had been drug-treated and did not have the opportunity to access the parasites.*

Q.10. Since subjects received different numbers of infectious mosquito bites, a table should have been included for subjects in the control and vaccine groups documenting exactly how many infectious mosquito bites were used. The force of infection is a determinant of likelihood of infection in CHMI.

R.10. *Table 2, page 28 was modified. A column with the number of infected mosquito bites per volunteer, as well as an index of potential number of sporozoites/bite, was included. Also Methods section on page 9, lines 162.*

Volunteers were exposed to 2-4 infected mosquito bites confirmed by mosquito dissection after the CHMI. We have found here and in previous CHMI that there is no difference in prepatent period or parasitemia density, and the number of infective bites, even in a range between 2 and 10 infected mosquito bites (ref 19).

Q.11. The discussion would benefit from 1) how antibody or CMI (IFN-g) play a role in protection; 2) what modifications may be necessary to improve vaccine efficacy (modification to vivax sequences or alternative adjuvants?); differences in naive subjects versus semi-immune subjects related to challenge outcome and immune responses

R.11. *This has been expanded in the Discussion section, page 16, lines 323-331. We do not envisage any modification; instead, we plan to conduct a larger trial to improve the power of the studies and reduce the heterogeneity of the volunteers.*

Reviewer #2

These are complex studies and important to conduct. Promising results are shown but there are many questions regarding the trial:

Q.1 -could not find the registration of this particular trial

R.1. *ClinicalTrials.gov number: **NCT 02083068** was in the front page since submission.*

Q.2. -the recruitment started in 2014, when was it completed?

R.2. *As described in the Study Design and Participants section, page 6, line 96. Volunteers' recruitment started on 03 October 2014 (first patient in) and was completed on 22 December 2014 (last patient in).*

Q.3 -when were the antibody and T cell assays performed ?

R.3. *T- cell studies were performed on the day of the volunteers' bleeding.*

Q.4. -the aim was to recruit 52 subjects but ended up recruiting 35, some explanation would be helpful.

R.4. *This process was better explained in Statistical Analysis, page 10, lines 188-193 and in Results section, page 11, lines 205-210.*

As described in the recruitment process, 121 volunteers attended the invitation to participate. We intended to identify 52 potential participants fulfilling the inclusion criteria and enroll 36. The inclusion criteria included several conditions that made enrolment challenging in an endemic population (Appendix 3). Unfortunately, 80/121 did not meet the inclusion criteria and had to be excluded. In addition, six declined their participation before enrolment; thus, we decided to start with the 35 available to avoid delays and greater losses.

Q.5. -how were the 121 subjects asked to participate, how and where were they asked and how many were from the endemic area and how many from non-endemic.

R.5. *This process was further explained in Section Volunteers enrolment and retention, page 11, lines 205-208.*

The volunteers were invited to participate by different means, including posters, flyers, radio, and TV broadcasting, followed by meetings and workshops to explain the study's objectives, risks, and benefits. However, in contrast to the other five clinical trials previously conducted by our group, this trial involved a longer duration. In addition, volunteers from the endemic region demanded to stay abroad for at least two months.

Volunteers were simultaneously recruited in Cali (phase IIa) and Buenaventura (phase IIb). Thirty-eight volunteers were initially from malaria-free areas and 83 from endemic sites (See Figure 1, Flow diagram, page 22).

Q.6. -then in the trial flow diagram we see 80 did not meet the inclusion criteria, which again raises the question how was the recruitment organized.

R.6. *This question is answered to this reviewer in Responses 4 and 5 above. This strengthens the challenge of these trials in endemic communities, even with the experience of several previous clinical trials in the same region (refs 16-19, 21).*

Q.7. -the flow diagram should indicate how many from endemic and non-endemic area were excluded

R.7. *This recommendation has been attended in the flow diagram and responded in R.5 (reviewer #2).*

Q.8. -detail of the recruited subjects should be given in more detail and what is meant by "This study sample is not representative of the whole population of individuals living in the country's malaria-endemic areas;"

R.8. *The population of Cali is mainly from the mestizo ethnic background (75%) and ~25 Afrodescendent, indigenous, and other minorities. The people from Buenaventura are about 90% Afro-descendent, with mestizo and indigenous minorities. For phase*

Ila, the participants were all mestizo (100%). For phase IIb, the participants were mestizo 11/16, afro-descendants and indigenous in a proportion that did not represent the population distribution in both sites.

Points regarding results:

Q.9 -figures can be improved considerably

R. 9. All figures were improved.

Q.10 -the antibodies shown are of those that react? the comparisons can be better presented.

R. 10. All participants seroconverted; therefore, the antibody titers shown here correspond to all reactive participants.

Q.11 -would be informative to show ELISPOT results for each individual (dot plots?)

R.11. ELIspot IFN- γ results were transformed in dot plots shown in Figure 4. Page 25 .

Abstract and Discussion:

Q.11 -not very precise when it comes to immunological reactivity

R. 11. We described in the original version that antibody reactivity and IFN- γ responses appear upon first immunization. Also that responded to each participant to the three different fragments of the PvCS protein after each immunization and CHMI using ELISA and ELIspot, respectively.

Q.12 -no discussion of the discrepancy between immunogenicity and protection

*R. 12. We disagree that the data presented have a discrepancy. On the contrary, they highly reproduce findings observed with other *P.falciparum* malaria pre-erythrocytic vaccines (ref 10, 11). See also responses to Q.3 and Q.4. from reviewer # 1.*

Nevertheless, we have extended these results in the Discussion, page 15 and 16, lines 314-336

Q.13 -deeper discussion of the findings in relation to data from other studies - patterns of parasite development, the immunological data

R. 13. Discussion of our findings was expanded in response to reviewer #1, in the Discussion section, page 16, lines 332-336, and page 18, lines 364-368.

On the other hand, regarding the “pattern of parasite development,” the study protocol indicated immediate treatment of the infection upon confirmation in blood circulation (patent parasitemia); then, it is impossible to determine its parasite development pattern beyond treatment. In this sense, the survival curve presented in Figure # 5 corresponds to the appearance of the parasite in circulation in those volunteers who were not fully protected.

Reviewer #3 (Remarks to the Author):

The authors present a *P. vivax* CHMI vaccine study demonstrating a level of sterilizing protection in trial participants, this seems of high interest to the study of malaria vaccines. The general message of the paper is concise and clear. My main comments focus on the statistical analyses.

Major comments:

I would recommend the authors have a statistician, preferably a co-author, review this manuscript to ensure that the analysis of the results is appropriate. I do not believe there are any irreparable deficiencies in the data and subsequent results, but I highlight several concerns in the statistical analysis below.

Issues related to protection efficacy (PE) statistics and subsequent reporting.

Q.1- The PE for the semi-immune group needs to account for the infection rate in their control group (See <https://doi.org/10.1093/infdis/jiaa421> as an example).

R.1- *We defined the infection rate taking into account the protection cutoff of the Phase IIa study and the analysis is explained in lines 287-290. Specifically, on line 288, we stated that “in the control group (n=5) three volunteers were considered protected (infection rate 3/5 = 40%) based on the protection cutoff of the Phase IIa study.*

To be more explicit we suggest to insert this sentence (infection rate 3/5 = 40%) on line 288.

Q.2- The protected semi-immune control participant cannot simply be excluded from the PE estimate as an outlier.

R.2- *On line 287 we have stated “In the Ctrl group, a volunteer did not develop patent parasitemia, and two others (total 3/5) (60%, 95% CI 0.31-0.89) developed parasitemia levels below 80 parasites/mL”, which indeed may continue communicating the message that the volunteer was not taking into account as control.*

We now suggest the following sentence (line 289)

“Using the protection cutoff of the Phase IIa study, three volunteers of the semi-immune Ctrl group were considered protected as parasitemia levels were below 80 parasites/L, including the volunteer (CSI 572) that did not develop patent parasitemia”.

Q.3- There is no formal statistical analysis of PE. At the very least, what is the precision (e.g., 95% CI) of these estimates? (reference above also addresses this)

***R.3-** We appreciate this recommendation and it was revised following the reviewer suggested publications. The 95% CI data was included in the Abstract, page 2, line 11; and in Results section, Vaccine Efficacy, page 14 lines 276-293.*

Q.4 There appears to be two different estimates of sterilizing PE in this study. In the abstract, sterilizing PE is reported as 5/11 (note typo in Abstract and Results, should be 45%, not 42%) and 4/11 based on “no parasitemia”.

***R.4-** We are sorry for not being explicit enough. We have described corrected the protective efficacy throughout the MS. Specifically, we have revised the description of PE in Abstracts section, page 2, line 11; Results section, pag 14, lines 273-295; and Discussion section, page 15, line 300-301.*

Revise crude data

Q.5-This result is repeated in Results section “Vaccine efficacy”, but a second set of numbers are also presented “based on the survival analysis”: 4/11 and 3/11.

The latter is then repeated in the lead Discussion paragraph. Unless there are multiple definitions of protection (e.g., sterilizing vs. reduction in parasitemia), there should only be one estimate of PE reported. If there were multiple definitions of protection, they should be more clearly defined and easily referenced.

***R.5-** It was revised throughout the MS as described in R.2 reviewer #3 and in previous responses.*

Q.6 - Lines 202-206 (Stats Methods), description of sample size calculation is unclear. What endpoint comparison is being powered?

***R.6-** This description of sample size calculation was extended in Statistical analysis section, on pag 10, line 188-193*

Q.7 What is the range of effect sizes?

***R.7-** Ranges of effects sizes were described wherever needed, throughout the Results section.*

Q.8 - Lines 206: "...protection efficacy was assessed at a 5% significance level and 80% power." Is this related to the sample size calculation?

R.8- *Yes, the significance was based on the sample size. We have revised the description of the sample size calculation, and have included 5% significance with 95% CI.*

Q.9 What comparison (ie, effect size) was assessed for 80% power? Why aren't there any formal comparisons of efficacy performed in this manuscript?

R.9- *Formal comparison of protective efficacy between groups as well as its association with immune responses were revised described in the Results section, Vaccine efficacy, page 14, line 276-295.*

Q.10- Comparison of immunological endpoints: it is unclear what statistical comparisons were performed and what criteria were used to determine their inclusion in the Results.

R.10- *The immunological endpoints were written in a more explicit way (Page 10 Statistical analysis, line 188) We compared production of specific antibodies and IFN γ in volunteers of the two phases IIa/IIb as well as between Exp and Ctrl groups (Results section, Page 12, line 240-264).*

Q.11 All performed tests should be clearly reported so the reader has a scope of the total hypotheses being evaluated. Generally, I was confused whether p-values represented tests over time (times often not clearly stated) or between cohorts; and there were no tables providing this information.

R.11- *We have tried to be as clear as possible to facilitate the comprehension of the study. p values were calculated between cohorts.*

Q.12 - There are responder definitions for both assays. It is clearly stated that all vaccinated participants were seropositive via ELISA post-all vaccinations. Is this true for the ELISPOT responses? Can a similar sentence lead those results?

R.12- *The recommended sentence about ELISPOT responses was included in the Results section, page 13, line 254. IFN- γ production was variable and at different time points there were negative*

Q.13 In addition to the binary protection-derived endpoints--sterilizing protection and delay in patency--can we learn anything from the qPCR continuous readout? For example, does parasitemia look similar among infected vaccinated and control participants? Correlations between immunological and parasite endpoints are generally of interest, but perhaps that is a follow-up analysis.- <https://doi.org/10.1371/journal.pcbi.1005255> is a theoretical overview of parasitemia assessment.

R.13- *The qPCR was performed retrospectively. It was found that in both, control and experimental semi-immune non-protected subjects parasites were detected 2-3*

days before in comparison to microscopy detection. However, in the case of naïve group no differences was observed on the day of parasite detection by the two methods.

In addition, apart of the sterily protected volunteer all others displayed highly variable parasite densities, ranging from 20-400 parasite/ μ L by microscopy (see Table 2, page 28). We did not find correlation between antibody titers and microscopic parasitemia-frequency or density, neither at group or individual level, nor with IFN- γ or the association of both.

Minor comments

Q.14 - The authors clearly have a great understanding of the evolving vaccine technology, but the introduction would benefit if it had a little more clinical/epidemiology background.

R. 14- *The description of P.vivax clinical manifestations and epidemiology was expanded in page 3, lines 23- 33.*

Q.15. How is PE assessed throughout studies, linking PE between field and CHMI studies, etc.. For example, [doi: 10.1016/j.pt.2016.11.001] appears to be a relevant review including description of the authors' past efforts.

R. 15- *We find this statistical calculation and models highly valuable for malaria vaccine efficacy. However, case of P. vivax CHMI analysis deserves the analysis and report in a separate publication.*

Q.16. - Figure 2: what do the square points denote in the plot?.

R. 16- *Square points are the symbols of Exp or Ctrl volunteers*

Q.17. - Figure 3:what do the shapes denote? Horizontal dashed lines? Would be a nice if the vaccine times were visualized or stated in the caption.

R. 17- *Figure 3, (now figure 5) page 26 was improved, and vaccination as well as CHMI time-points were inserted. Each symbol indicates and individual volunteers (Naïve and Semi-immune). Dashed lines were eliminated to avoid confusion.*

Q.18. - Figure 4: light spaghetti behind boxes may help highlight the trends. Unclear if CHMI box is all zero or smushed by scaling, log-transforming could help. Consider denoting positive responders.

R. 18- *Figure 4 (page 25) was changed as recommended.*

Q.19 - Figures 3-4: Helpful to state cutoff value.

R. 19- *Cutoff values do not apply in Figure 3 because they are subtracted to calculate the antibodies RI reactivity. In Figure 4, cut off value was inserted (line 475).*

Reviewers' Comments:

Reviewer #2:

Remarks to the Author:

The authors have improved clarity by providing more information regarding the trial, the timing, the set up and immunological data. However, I would really urge them to work more the figure legends so that the readers can get a good idea of the data. Just one example, in Figure 4, we have no information what we are looking at, tell us please more about the figure. In the text before referring to figure 4 you mention and controls were negative, where is the data? the number of subjects in the antibody data fig 3 and 4 do not match it seems but why not indicate how many are we supposed to see in the legends. What is 0 in fig 4 ? background or after first immunization? as you state there are three immunization and describe the boosting, should there not be lines to show the boosting in figure 4? if zero is the first immunization what is the background ELISPOT result in naive and semi immunes. Would it not be interesting to show those with sterile immunity with a special symbol?

Reviewer #3:

Remarks to the Author:

The authors addressed my comments regarding vaccine efficacy. I had several follow-up comments and questions around the immunogenicity analysis that are generally limited to the revised text. With the exception of my comment #2 below on the regression analysis, I believe most of these comments can be addressed with minor writing edits.

1. The authors added correlates results (association between immune response and protection), a nice addition to the paper even if the associations were not significant (Ex. Lines 281-281).

- What tests were used to assess these associations? Was parasitemia treated as binary or continuous?

- Do the authors think that the insignificant results are due to lack of precision (low sample size, high response variation) or were the average immune response magnitudes relatively similar between protected and infected participants?

2. From methods (line 218): Titers and IFN-g production were compared "among groups at several time points using regression analysis". This is not reproducible as written. Can the authors elaborate on their regression analysis? Are there separate models for each fragment? If time is in the model, are they controlling for repeated measures (ex, mixed effects model) so the p-values are correctly specified?

3. In the following lines, it was unclear what was being tested:

- lines 274-275: A set of p-values by fragment is provided for each study group. What is being compared here?

- lines 270, 276-278 and Fig 4 caption: these p-values are cohort comparisons for each fragment, but at which time point?

- lines 284-286: these appear to be tests of decreasing response between final vaccine time and CHMI; but in R.11 to Reviewer #3's original comments the authors state "p-values are between cohort" so that is unclear to me. There are no stats methods stated for comparisons over time (e.g., paired tests).

- line 348: reference to decrease in IFN-g for the naive group appears similar to the result stated in line 285, but p-values are different.

4. Assuming I had the correct set of figures (5 total), there were some cases in the revised text where it appears the incorrect figure was referenced:

- Line 280: IFN-g result but references survival fig (Fig 5)

- line 306: PE results references antibody fig (Figure 3)

- line 319: comparing prepatent periods references antibody fig (Figure 3)

5. Lines 216-221 and lines 221-226 are repeating the same text.

Rebuttal

Reviewer #1

Question/Comment #1

Regarding your response to the concerns of reviewer #1, please provide a citation for the grade of severity of AEs (the currently provided link in reference 29 doesn't seem to provide the necessary information).

Response #1

Reference 29, currently refs #35, was changed for another more specific, mentioned in line 246.

Q/C # 2

Include the additional information you provided on the subject with recurrence of *P. vivax* malaria in *the manuscript*.

R #2. The information on the subject with P. vivax recurrence was reinserted in the manuscript in the Results and Discussion sections:

In the Results section, Vaccine Efficacy, line 320, the following sentence was added: "Unexpectedly, one of the semi-immune Ctrl group participants developed parasitemia (SAE) about two months after returning to the endemic areas, probably due to reinfection."

In the Discussion section, lines 349-52, the following sentence: "Regarding the infection experienced by one of the semi-immune Ctrl participants after returning to the endemic area, it may have corresponded to reinfection. Unfortunately, the volunteer was in a distant rural setting, and the parasites were not available for genotyping; this patient was one of the susceptible volunteers (not protected)."

REVIEWER COMMENTS

Reviewer #2 (Remarks to the Author):

Q 3. The authors have improved clarity by providing more information regarding the trial, the timing, the set up and immunological data. However, I would really urge them to work more the figure legends so that the readers can get a good idea of the data. Just one example, in Figure 4, we have no information what we are looking at, tell us please more about the figure.

R 3. The suggested changes were made in the figures, and the legends of all figures and tables were extended to make them more explicit.

In figure 3, we included **a)** antibody titers of control volunteers, in both naïve and semi-immune groups; **b)** the number of volunteers per antigen (N, R, C) group were inserted in the figure and described in the figure legend; **d)** volunteers who developed sterile immunity were indicated with dashed lines.

Likewise, in the main body of the MS, in the vaccine immunogenicity section, pages 12 and 13 (lines 263-289), we inserted several sentences to clarify it.

Just one example, in Figure 4, we have no information what we are looking at, tell us please more about the figure.

In the previous version, we had mixed up the numbers of some of the figures. Because of this, the reviewer interpreted the sentence as referring to IFN- γ . We have corrected the figures' numbers. In addition, in Figure 4, we included the IFN- γ levels of the negative control and explained these changes both in the Figure legend and the text, as follows:

“Figure 4. Depicts the single-cell IFN- γ *ex vivo* production by fresh PBMC from naïve and semi-immune volunteers collected before immunization (times 0, 1, 2) and CHMI. Values are expressed as IFN- γ -sfc/ 10^6 in response to 40 h of *in vitro* stimulation with each PvCS protein fragment (N, R, C). Cells produced IFN- γ upon stimulation with the different fragments in an unstable manner throughout the study phases, without a clear boosting trend. Neither naïve nor the semi-immune IFN- γ levels were associated with parasitemia. Red symbols denote protected volunteers.”

Q 4: In the text before referring to figure 4 you mention and controls were negative, where is the data?

R 4: See previous response

Q 5: the number of subjects in the antibody data fig 3 and 4 do not match it seems but why not indicate how many are we supposed to see in the legends.

R 2: As suggested, we revised the texts for coherence and inserted the number of subjects in each figure. See responses R3 and R4.

Q 6: What is 0 in fig 4, background or after first immunization? as you state there are three immunization and describe the boosting, should there not be lines to show the boosting in figure 4?

R 6: *The value at time 0 in figure 4 corresponds to the pre-immunization bleeding (first vaccine dose), and the other bleedings marked as with the numbers 1 and 2 correspond to the IFN- γ levels determined one month after immunization #1 and three months after immunization #2. These bleedings 1, 2, and 3 were performed before immunization 1, 2 and 3.*

IFN- γ level in figure # 4 corresponds to the value obtained after deducting the background value (cells without peptide stimulation) from the total value (cell after peptide stimulation). Regarding the boosting effect was described in the legend and the main body lines 196 - 199.

Q 7: If zero is the first immunization what is the background ELISPOT result in naive and semi immunes.

R 7: *Explained in the R 6*

Q 8: Would it not be interesting to show those with sterile immunity with a special symbol?

R 8: *Volunteers who develop sterile immunity were indicated with dashed curves in Figures 3, and in Figure 4 used red color symbols.*

Reviewer #3 (Remarks to the Author):

The authors addressed my comments regarding vaccine efficacy. I had several follow-up comments and questions around the immunogenicity analysis that are generally limited to the revised text. With the exception of my comment #2 below on the regression analysis, I believe most of these comments can be addressed with minor writing edits.

Comment 1. The authors added correlates results (association between immune response and protection), a nice addition to the paper even if the associations were not significant (Ex. Lines 281-282).

Q 1- What tests were used to assess these associations? Was parasitemia treated as binary or continuous?

R 1. *The association between immune response and protection was assessed using paired sample t-test, and parasitemia was treated as a continuous variable. It was also added in the Statistical analysis section, on page 11, lines 221-224.*

Q 2. - Do the authors think that the insignificant results are due to lack of precision (low sample size, high response variation) or were the average immune response magnitudes relatively similar between protected and infected participants?

R 2: We consider that the lack of significance is probably due to the small sample size and the highly variable immune response among participants, likely due to the population heterogeneity in terms of ethnicity, sex and nutrition status, among other potential factors. This was also included in the main body of the MS, in the Discussion section on pages 17-18, lines 360-363.

Q 3. From methods (line 218): Titers and IFN-g production were compared "among groups at several time points using regression analysis". This is not reproducible as written. Can the authors elaborate on their regression analysis? Are there separate models for each fragment?. If time is in the model, are they controlling for repeated measures (ex, mixed effects model) so the p-values are correctly specified?

R 3. The regression analysis was performed both as separate models per fragment (N, R, and C) and grouped (N+R+C); both analyses indicated no correlation with protection.

IFN-g titers (levels) were compared using the mean value obtained in the last two volunteers' bleedings (at months 7th and 10th), which were the closest bleedings to the CHMI time and expected to be more associated with protection. At this time, the IFN- γ analysis was performed using the three fragments together (N+R+C).

We consider that the p-values are reliable. Although we did not control for repeated measures (ex, mixed effects model), we consider that the homogeneity of the system (i.e. human subjects, type of samples, lab methods) leads to reliable reproducibility of the used in the model. In each individual assay (at different time points), IFN- γ background levels released by PBMC without peptide stimulations were determined and subtracted from the values obtained from the values produced by PBMC cultures with peptide stimulation.

The following text was included on page 10, lines 196-199.

“In each assay, PBMC were also incubated without peptides to determine the IFN- γ background production level. The background level of each test was subtracted from the specific counts obtained from peptide stimulated cells.”

Q 4. In the following lines, it was unclear what was being tested: - lines 274-275: A set of p-values by fragment is provided for each study group. What is being compared here?

R 4: As described in R3, the referred text compared the IFN- γ levels at months 7th and 10th, corresponding to time-2 and CHMI of Figure 4. This text is currently in lines 281-282, as follows:

“Nevertheless, neither the naïve (N, $p=0.67$; C, $p=0.94$; R, $p=0.15$) nor the semi-immune IFN- γ levels (N, $p=0.76$; C, $p=0.71$; R, $p=0.48$) determined on month 7th before CHMI, were associated with parasitemia”.

Q 5- lines 270, 276-278 and Fig 4 caption: these p-values are cohort comparisons for each fragment, but at which time point?

R 5: As mentioned in the R 4 response, the p values correspond to IFN- γ levels at time-2 and CHMI time shown in Figure 4.

Q 6- lines 284-286: these appear to be tests of decreasing response between final vaccine time and CHMI; original comments the authors state "p-values are between cohort" so that is unclear to me. There are no stats methods stated for comparisons over time (e.g., paired tests).

R 6: As mentioned in R1 and R5, to make this text more clear and precise, we specified that the association between immune response and protection had been assessed using paired sample t-test, taking into account the IFN- γ values obtained from cells collected on months 7th and 10th (time-2 and CHMI time). Parasitemia was treated as a continuous variable.

It is highlighted in the Statistical analysis section, on page 11, lines 221-224.

Q 7- line 348: reference to decrease in IFN-g for the naive group appears similar to the result stated in line 285, but p-values are different.

R 7: The reviewer is correct; we should have written (N-peptide, $p=0.0022$, R-, $p<0.0003$, and C-, $p=0.0367$). It has been corrected in the vaccine immunogenicity section, in line 286.

Q 8- Assuming I had the correct set of figures (5 total), there were some cases in the revised text where it appears the incorrect figure was referenced:

- Line 280: IFN-g result but references survival fig (Fig 5)
- line 306: PE results references antibody fig (Figure 3)
- line 319: comparing prepatent periods references antibody fig (Figure 3)

R 8: The reviewer is correct; we had mixed up the sequence of figures in some parts of the MS. Figure numbers were revised throughout the text.

Q 9. Lines 216-221 and lines 221-226 are repeating the same text.

R 8: We revised the whole text again to avoid any text duplications

Finally, apart from the specific changes in response to the reviewers' comments, we have made minor but essential edits to make the text more explicit, all of which are highlighted in blue.

Lines 37, 113, 146, 181, 184-187, 190, 211, 214, 264, 269, 336.

However, none of them changes the essence of the MS.

Reviewers' Comments:

Reviewer #3:

Remarks to the Author:

The authors have addressed my comments and I have no further comments.